# Quantitative and Qualitative Running Gait Analysis through an Innovative Video-Based Approach

**DOI:** 10.3390/s21092977

**Published:** 2021-04-23

**Authors:** Laura Simoni, Alessandra Scarton, Claudio Macchi, Federico Gori, Guido Pasquini, Silvia Pogliaghi

**Affiliations:** 1Department of Neurosciences, Biomedicine and Movement Sciences, University of Verona, 37129 Verona, Italy; laura.simoni@univr.it (L.S.); silvia.pogliaghi@univr.it (S.P.); 2IRCCS Fondazione Don Carlo Gnocchi ONLUS, 50143 Florence, Italy; claudio.macchi@unifi.it; 3Microgate SRL, 39100 Bolzano, Italy; alessandra.scarton@microgate.it (A.S.); federico.gori@microgate.it (F.G.)

**Keywords:** gait analysis, treadmill running, video-based systems, harmony, fast Fourier transform, video sensors

## Abstract

Quantitative and qualitative running gait analysis allows the early identification and the longitudinal monitoring of gait abnormalities linked to running-related injuries. A promising calibration- and marker-less video sensor-based technology (i.e., *Graal*), recently validated for walking gait, may also offer a time- and cost-efficient alternative to the gold-standard methods for running. This study aim was to ascertain the validity of an improved version of Graal for quantitative and qualitative analysis of running. In 33 healthy recreational runners (mean age 41 years), treadmill running at self-selected submaximal speed was simultaneously evaluated by a validated photosensor system (i.e., *Optogait*—the reference methodology) and by the video analysis of a posterior 30-fps video of the runner through the optimized version of Graal. Graal is video analysis software that provides a spectral analysis of the brightness over time for each pixel of the video, in order to identify its frequency contents. The two main frequencies of variation of the pixel’s brightness (i.e., F1 and F2) correspond to the two most important frequencies of gait (i.e., stride frequency and cadence). The Optogait system recorded step length, cadence, and its variability (vCAD, a traditional index of gait quality). Graal provided a direct measurement of F2 (reflecting cadence), an indirect measure of step length, and two indexes of global gait quality (harmony and synchrony index). The correspondence between quantitative indexes (Cadence vs. F2 and step length vs. Graal step length) was tested via paired t-test, correlations, and Bland–Altman plots. The relationship between qualitative indexes (vCAD vs. Harmony and Synchrony Index) was investigated by correlation analysis. Cadence and step length were, respectively, not significantly different from and highly correlated with F2 (1.41 Hz ± 0.09 Hz vs. 1.42 Hz ± 0.08 Hz, *p* = 0.25, r^2^ = 0.81) and Graal step length (104.70 cm ± 013.27 cm vs. 107.56 cm ± 13.67 cm, *p* = 0.55, r^2^ = 0.98). Bland–Altman tests confirmed a non-significant bias and small imprecision between methods for both parameters. The vCAD was 1.84% ± 0.66%, and it was significantly correlated with neither the Harmony nor the Synchrony Index (0.21 ± 0.03, *p* = 0.92, r^2^ = 0.00038; 0.21 ± 0.96, *p* = 0.87, r^2^ = 0.00122). These findings confirm the validity of the optimized version of Graal for the measurement of quantitative indexes of gait. Hence, Graal constitutes an extremely time- and cost-efficient tool suitable for quantitative analysis of running. However, its validity for qualitative running gait analysis remains inconclusive and will require further evaluation in a wider range of absolute and relative running intensities in different individuals.

## 1. Introduction

Running is one of the most popular recreational physical activities in the world, as it provides substantial health benefits at minimal expense [1,2]. However, together with numerous health benefits, there is a relatively high incidence of running-related injuries (6.8 to 59 injuries per 1000 h of running) [3,4]. The etiology of running-related injuries is multifactorial and still not fully understood [5]. Quantitative (e.g., stride length, cadence) and qualitative (e.g., gait variability, gait harmony) biomechanical features are modifiable risk factors that play a relevant role in the occurrence of injuries, particularly in recreational and novice runners [6,7]. Quantitative gait analysis measures the spatiotemporal parameters, kinematics, and kinetics of running. Spatiotemporal parameters (e.g., cadence, stride length, and contact and flight times) describe the basic features of gait pattern and are useful to easily defining the ability of the individual to fulfil the general requirements of running (e.g., symmetry, coordination, and gait economy) [8,9,10]. Alteration in these parameters, such as the presence of overstride (i.e., an excessive stride length associated with a decreased cadence) or an excessive cadence variability, contribute to increased risk of injuries and to decreased running economy. Moreover, the longitudinal monitoring of these parameters allows the follow-up of functional outcomes after rehabilitation treatments [11,12,13]. Qualitative gait analysis identifies specific running features (e.g., foot strike pattern, presence of cross-over, and low gait harmony) that are associated with increased risk of overuse injuries [13] or global movement scores (e.g., Volodalen Scale) indicative of running economy and inversely related to the risk of overuse injuries [14].

The quantitative and qualitative analysis of running biomechanics can be performed in either a laboratory or an outdoor setting, depending on the aim and on the available equipment. Running gait analysis in the laboratory setting has the main advantages of fully controlled environmental conditions and the use of gold-standard methods (i.e., optical motion capture systems associated with force platforms). Limitations of the use of gold-standard methods are a not fully ecological walking or running style, due to the use of a treadmill, and the high costs and complexity in terms of experimental setup [15,16,17,18]. In recent years, more low-cost, easy-to-use, and marker-less alternatives to gold-standard methods have been developed for large-scale gait evaluation [6,8,13,14]. The most frequently used alternative technologies for quantitative gait analysis are optical timing systems and inertial sensors. Despite the several advantages compared to gold-standard methods, these systems require setup/subject preparation, calibration, and data post-processing procedures that reduce the ease of use. In recent years, no-calibration and marker-less video analysis methods have been developed, but their accuracy and precision remain to be assessed by validation studies [19,20,21].

Qualitative analysis of running is traditionally performed by clinicians or expert coaches using evaluation scales during direct observation of gait or offline video analysis [13,14,22]. Objective methods have been proposed to overcome the limitations of the above subjective approach, the most frequently used alternative being inertial sensors. Through the study of the biomechanics of separate body segments that represent the entire body and/or through the evaluation of the interaction, coordination, and symmetry of multiple body segments, these techniques permit a simple evaluation of gait quality [6,23,24,25,26,27]. Traditionally, running gait harmony is evaluated by considering the variability of spatiotemporal gait parameters, left vs. right asymmetries, and the step-by-step rhythmicity of acceleration patterns of the center of mass (i.e., the harmonic ratio) [6,28,29,30,31].

The above cited traditional and alternative methods for the quantitative and qualitative evaluation of gait all have in common a focus on specific body segments (e.g., center of mass, lower limbs, or lower and upper trunk) that are considered representative of the motion of the entire body [5,28,32,33,34]. However, taking into account that all forms of gait are characterized by a simultaneous movement of all the lower and upper body segments, recent studies have shown that a global approach to movement characterization can provide a synthetic view of the harmony and quality of gait [19,32,35] while considerably reducing the cost and time of gait analysis. Low-cost and calibration-/marker-less technologies would allow gait evaluation on a large scale in both clinical and sport sciences towards early identification of injury risk, injury prevention, and monitoring of recovery for a swift and safe return to sport. In the context of sport coaching, the same technologies would provide indexes that are objective, accurate, and repeatable to describe and monitor running technique for performance enhancement. However, a methodology providing quantitative and qualitative indexes of running gait with a global approach is still missing.

With the aim to provide a tool for large-scale gait analysis, we recently proposed an innovative, video-based method that relies on the analysis of the frequency content of the variations of pixel brightness in digital video of a cyclic movement [35]. This method, validated for the evaluation of physiological walking at self-selected speed, is able to extract quantitative parameters such as the main frequency contents of rhythmic movement (i.e., cadence and step frequency), along with a qualitative index of the harmony of gait [35]. While the above findings are very promising, some technical optimizations and additional confirmatory studies are needed before the technique can be extensively applied to the study of human movement. One of the main limitations of the method was the complexity of the algorithm used to extract the main frequencies of interest and, thus, the time needed for the computation. Therefore, in this study we optimized the calculations by reducing the signal noise and by averaging the frequency content of the video into a single spectrum. Furthermore, in search of a possibly more informative index of gait quality, we explored the performance of an alternative index of dispersion/variability of the average spectrum.

This study tested the hypothesis that the optimized version of this innovative video-based method for gait analysis is able to accurately extract quantitative parameters and indexes of gait quality. The method’s performance in terms of the quantitative and qualitative evaluation of gait was tested on running, a form of locomotion that is characterized by a higher frequency and complexity compared to walking [32].

## 2. Materials and Methods

In the present work, running data from 33 healthy individuals (21 males, 12 females) were acquired and studied. Participants signed a written informed consent, and the study was approved by the ethical committee of the IRCCS Fondazione Don Carlo Gnocchi ONLUS (number 13663_oss). Runners had to be engaged in a running program (at least two sessions per week with a minimum continuous running time of 20 min per session) for a minimum of six months, be free from injuries in the last two months, be free from chronic musculoskeletal diseases, and be adapted in the use of a treadmill [36].

Subjects were requested to run on a flat motorized treadmill (MTC-Climb, Runner, Italy) following a previously developed protocol [37]. In brief, subjects started with five minutes of a warm-up and familiarization phase: the treadmill velocity was initially set at 7 km/h and was self-modulated until a subjectively comfortable speed was reached [38,39]. The subjectively comfortable speed was chosen to minimize the fatigue factor, which has been shown to affect running mechanics [36]. All participants wore their habitual running shoes. Following warm-up/familiarization, after a 1-minute recovery, subjects ran at the predetermined self-selected speed for 2 min. The analysis of running parameters was performed on the final 60” of the running test at self-selected speed, in parallel, through a validated photosensor system (Optogait) [40] that was used as a reference method and through custom-designed, Web-based video analysis software (Graal) [35]. Both Optogait and Graal are gait analysis systems developed by the company Microgate (Bolzano, Italy).

Among the parameters measured directly by the Optogait system, we extracted step length (cm) and step length normalized for stature, contact time (s), flight time (s), cadence (in Hz), and within-subject cadence variability across the entire test (vCAD, %) [7]. The latter was taken as an index of gait harmony [27,30,41,42].

Concurrently, using a Logitech Brio 4 K (2.5 m behind the subject at 95 cm height), we recorded videos at 30 Hz and with a resolution of 640 × 480 pixels that were successively analyzed by Graal software. The original method of analysis, optimized in the present work, was previously reported in detail [35]. In brief, the following steps were performed via MATLAB-based software (The MathWorks, USA, version R2018) in order to identify the main frequency content of gait (i.e., F2, in Hz, representative of cadence), to derive the step length (Graal step length, in cm), and to calculate an index of global gait quality (i.e., Harmony Index). In the present study an additional index (i.e., *Synchrony Index*) was developed and proposed to describe the global gait quality. Both the Harmony and Synchrony Indexes are in arbitrary units [35]. The analysis starts with the subdivision of the video file into frames, allowing us to obtain a number (n) of individual JPEG images. For each pixel of each image, the software computes the brightness via a brightness edit algorithm [35,43], followed by the application of a specific correction method called the Hanning window and a high-pass Butterworth filter with order 2 at 0.5 Hz to attenuate the component due to direct current [44]. After this preprocessing, the fast Fourier transform (FFT) algorithm used in Simoni et al. (2020) [35] is applied to the vector of brightness values over time for each pixel to obtain its magnitude in the frequency domain. The result is a power spectrum for each pixel that displays the magnitude as a function of frequency and permits us to identify the peaks of the dominant frequencies [35]. To ameliorate the process of peak frequency detection presented in Simoni et al. (2020), power spectra for each pixel are then averaged in order to obtain the Averaged Power Spectrum of the video, and the signal is then further processed using the detrend MATLAB function in order to attenuate the effect of flickering or pink noise [44] (Figure 1).

A simple peak detection is thus performed to extract the peaks with the highest power between 0 and 15 Hz, as indicated in Simoni et al. (2020) [35]. The frequencies of the first two largest peaks are named F1 and F2 and have been demonstrated to correspond, respectively, to stride frequency and cadence, expressed in Hz units, for walking gait [35].

Assuming that F2 corresponds to step frequency or cadence, the Graal system indirectly estimates the spatial parameter step length as follows [35,45,46]:(1)$$Graal\;Step\;Length\;\left[\mathrm{cm}\right]=speed\;\left[m\times s^{-1}\right]\times F2^{-1}\;\left[s^{-1}\right]\times 100$$ F2 is expressed in Hz, but in Equation (1), *Hz* have been represented as $s^{-1}$ to facilitate readability.

Moreover, in search of a more informative descriptor of global gait harmony, we computed the Synchrony Index, intended to restitute the degree of synchrony in the change of brightness of the pixels that compose the body image. The Synchrony Index is calculated through the ratio between the height and the width of the peak F2 of the Averaged Power Spectrum representing cadence (Figure 2).

A higher height-to-base ratio results in a higher Synchrony Index and indicates a narrow distribution of the frequencies around the peak. This is indicative of a high synchrony between pixels that should, in turn, represent a better quality of gait in terms of harmony and intra- and inter-segmental movement coordination and variability.

Finally, the Harmony Index was calculated as previously described [35]. In brief, the list of frequencies from all the pixels was ordered from the lowest to the highest and plotted over line number to create a frequency-distribution plot (Figure 3). The Harmony Index is calculated as the fitting error of a polynomial of degree 15 to this plot [35]. A high “stepiness” of the frequency-distribution plot (as in Figure 3, left panel) results in a larger fitting error, indicative of a lower variability in the gait pattern [35].

For all variables we calculated the average and ±SD (standard deviation) on a 60” duration of the test for the overall population and separately for males and females. Quantitative and qualitative running parameters obtained by the reference system (i.e., Optogait) were compared to those calculated through Graal. Visual inspection and the application of the Shapiro–Wilk test showed that all data were normally distributed. The paired t-test, Pearson product-moment correlation analysis, and 95% limits of agreement by Bland and Altman plots were used to test the correspondence between cadence and F2 and between step length and Graal step length. The association between the Harmony Index and Synchrony Index, and the associations of each of them with vCAD, were tested by Pearson product-moment correlation analysis. Data were analyzed using IBM SPSS Statistics Version 23.

## 3. Results

Subjects’ characteristics are reported in Table 1.

Subjects ran at a self-selected speed of 10.6 (0.5) km/h, with a step length of 108.12 (13.04) cm and a normalized step length of 0.62 (0.06). Values of staptiotemporal gait parameters measured with the reference methods have been reported extensively in Table 2.

Cadence and step length were not significantly different from and highly correlated with, respectively, F2 (1.41 Hz ± 0.09 Hz vs. 1.42 Hz ± 0.08 Hz, *p* = 0.25, r^2^ = 0.81) and Graal step length (104.70 cm ± 013.27 cm vs. 107.56 cm ± 13.67 cm, *p* = 0.55, r^2^ = 0.98) (Figure 4). Bland–Altman plots of cadence and step length against F2 and Graal step length, respectively, are presented in Figure 4 in the last panel. Bland–Altman tests confirmed a non-significant bias and small imprecision.

Measures of harmony derived from the video analysis method are reported in Table 3. The two indexes were not significantly correlated with each other (*p* = 0.268, r^2^ = 0.072) and neither the Harmony nor the Synchrony Index was significantly correlated with vCAD (Table 3).

## 4. Discussion

In this study, we developed an optimized version of Graal, an innovative video-based method for gait analysis, and evaluated its performance in identifying the main quantitative and qualitative characteristics of running gait compared to an optical timing-based method used as a reference. The results of this study confirm that the optimized version of Graal presented in this work can accurately provide the main frequency content (i.e., cadence) and spatial parameters (i.e., step length) in a form of locomotion that is characterized by a higher frequency and complexity compared to walking [32]. However, in the homogeneous, healthy recreational runners’ population tested in the study, the indexes of global gait quality derived from Graal analysis (both the previously described Harmony Index and the newly developed Synchrony Index) appear unrelated to gait variability.

The population studied is a representative sample of Italian expert recreational runners of both sexes in terms of training habits, level of training (see the inclusion criteria), age, and anthropometric characteristics [47]. Moreover, the values of spatiotemporal gait parameters measured by Optogait (i.e., the reference system used in the study) are consistent with those reported in previous studies that analyzed the running gait of recreational runners at similar self-selected speeds [8,21,26], thus confirming the results already present in the literature.

In agreement with a previous study from our group that demonstrated the accuracy and precision of Graal in identifying quantitative gait parameters during walking, the current study confirms the accurate detection of spatiotemporal gait parameters even for running at self-selected speed, a form of locomotion characterized by a higher frequency content and more complex inter- and intra-joint and body segment coordination patterns [32].

The possibility of estimating spatiotemporal parameters through the analysis of the frequency content of gait, without the need to identify the initial contact for each step of the video, as in the common video-based systems, is supported by the work of CJC Lamoth and colleagues. In their work, they applied principal component analysis (PCA) to the study of multisegmental coordination of body structures in walking and running, identifying the correspondence between the first component in frequency and the main frequency of locomotion (i.e., stride rate) [48]. Therefore, the above cited study provides the conceptual framework for the use of the frequency components of a cyclic activity to derive spatiotemporal gait indexes. The assumption on which our methodology relies is, in fact, the correspondence between the two main frequencies of variation of the pixel brightness in a video of walking/running gait and the two main frequencies of gait (e.g., stride frequency and cadence). However, the approach of CJC Lamoth and colleagues applied to the calculation of spatiotemporal gait parameters is seldom used in a clinical environment, because it does not provide additional information compared to the motion capture method on which it relies.

Other methods exist that aim at providing either a simpler or more cost-efficient approach to movement analysis, overcoming the main limitations of the gold-standard, optoelectronic motion-capture approach. Among them, methods based on 3D video plus specific software for the analysis eliminate the necessity of using markers (e.g., Microsoft Kinect). In addition, methods such as the visual analysis performed by a clinician of a simple 2D video [21], recorded via normal commercial/mobile phone cameras, further reduce costs. Unfortunately, the latter methodology is burdened by subjectivity (e.g., reliance on the experience of the clinician performing the video analysis), low external validity (e.g., the influence of the setup), and understudied overall validity [16,20]. Compared to the above solutions, Graal offers an automatic, non-operator-dependent system that provides robust performance independent from the absolute ambient light conditions (though it requires a stable, artificial light source) [35] and the camera’s sampling rate (though it requires a minimum of 30 Hz and a minimum of 2048 frames) and that can be used on videos with a low pixel resolution. The relative independence from sampling rate was verified in our study by comparing the analysis of the same videos (i.e., 30 s of running at self-selected speed) sampled at 30 vs. 60 Hz in an independent sample of five adult runners. The main frequencies derived from the 30 Hz files were highly correlated with and not significantly different to those from the 60 Hz files (F1 = 1.45 Hz ± 0.08 Hz vs. 1.45 Hz ± 0.08 Hz, *p* = 0.86, r^2^ = 0.99, F2 = 2.91 Hz ± 0.15 Hz vs. 2.90 Hz ± 0.15 Hz, *p* = 0.26, r^2^ = 0.99). With regard to the camera resolution, while a 4K camera was used in our study, we decided to analyze videos at a lower resolution (i.e., 640 × 480 px); this decision was the result of a compromise between Graal practical considerations and accuracy. In fact, the 640 × 480 px resolution allows us to describe the full human figure with a sufficiently large number of pixels to characterize the dynamics of the movement (in the videos, the person’s silhouette occupied about 1/3 of the overall available space, equal to 102,400 pixels out of the overall 307,200 pixels). The use of the full 4K resolution potential of the camera would not add information and would raise the processing time from a few minutes to several hours, because an FFT needs to be performed on each pixel of the image. Moreover, 4K resolution videos usually have large file size, making it more difficult for a generic user to upload the video without a powerful Internet connection and to store it in an online database. Finally, the 640 × 480 resolution can be easily obtained through low-cost video equipment (e.g., mobile phones).

Together with quantitative analysis, Graal was proposed as an objective method to perform a global qualitative analysis of running through the use of the innovative indexes Harmony Index and Synchrony Index, which should reflect gait quality in terms of harmony, synchrony, and inter- and intra-segmental coordination and variability. The approach used in this study was firstly adopted by Williams and Vicinanza, who proposed a method of global qualitative gait analysis that consists of the evaluation of the main frequency content of gait through spectral analysis of the movement of 22 markers placed bilaterally on both the lower and upper body [32]. The method is based on the intuition that normal gait is a nearly periodic signal and that anomalies usually disturb such periodicity [35]. For this reason, we hypothesized that our newly developed gait harmony indexes Harmony Index and Synchrony Index, which are intended to respectively represent the variability of the gait pattern and the “spread” of the frequency content of the whole-body movements, would be linked with vCAD. Actually, our results did not confirm the initial hypothesis: the Harmony and Synchrony Indexes were significantly correlated neither with each other nor with vCAD (i.e., the reference measure for gait harmony used in our study). vCAD, derived from spatiotemporal gait parameters, expresses the variability of cadence as a percentage of the average value; as such, this index is often used to indirectly evaluate the overall quality of gait (e.g., harmony, motor control health) [6,28,29,30,31]. We speculate that the low variability of vCAD in the population studied (3.82% ± 0.15%, typical of a fully physiological running, i.e., between 1 and 4%), due to the homogeneity of the sample in terms of running speed, non-injured status, and running experience (i.e., the main elements affecting vCAD) [7,12], may interfere with our ability to identify significant correlations among the vCAD, Harmony Index, and Synchrony Index. In the effort to understand whether this was one reason for our unexpected results, we ordered data based on ascending vCAD values and divided the sample into Lower vCAD (15 subjects, 2.66% ± 0.36%) and Higher vCAD (16 subjects, 4.75% ± 1.45%) groups that were compared by *t*-test. vCAD was significantly different between the groups (*p* = 0.001), and the Synchrony Index was higher in the Lower vCAD group (2.69% ± 1.11% vs. 1.99% ± 0.62%, *p* = 0.04); however, the Harmony Index did not differ significantly between groups. These results suggest the existence of an inverse relation between indexes of global gait quality and variability of cadence that in the present work may not yield a significant correlation due to the extremely narrow range of variability of vCAD. Further investigation should consider a population characterized by a larger variability in running cadence (e.g., novice runners) and/or with a vCAD outside the range of normality (i.e., >7%) (e.g., injured runners) or the effect of absolute or relative exercise intensity on movement quality in a given individual.

From a practical standpoint, one limitation of the approach tested in the current study is that, as of today, its use is confined to indoor treadmill locomotion and constant illumination conditions.

In summary, to the best of our knowledge, the video-based system Graal is the first semi-automatic, objective method that accurately identifies the basic spatiotemporal parameters of running gait (i.e., cadence and step length) with a minimum time investment (i.e., 1 min of video recording + 10 min for the analysis). The above can be obtained based on the determination of the frequency content of a rhythmic motion, without the necessity of relatively expensive hardware/software, calibration, markers or sensors on the person’s body or on the running surface, and manual post-processing video editing. For these reasons, Graal can be a useful gait analysis method for all kinds of scenarios where a basic gait analysis is needed (e.g., medical clinics and training or research centres). The validity of Graal’s Harmony and Synchrony Indexes as measures of global gait quality still needs further research to be confirmed. Further studies about gait quality in different running conditions (e.g., submaximal and maximal running intensities) and on different populations (e.g., novice runners to treadmill running, injured, etc.) could be useful to explore the ability of the Harmony and Synchrony Indexes to reflect the quality of gait.

## Figures and Tables

**Figure 1 sensors-21-02977-f001:**
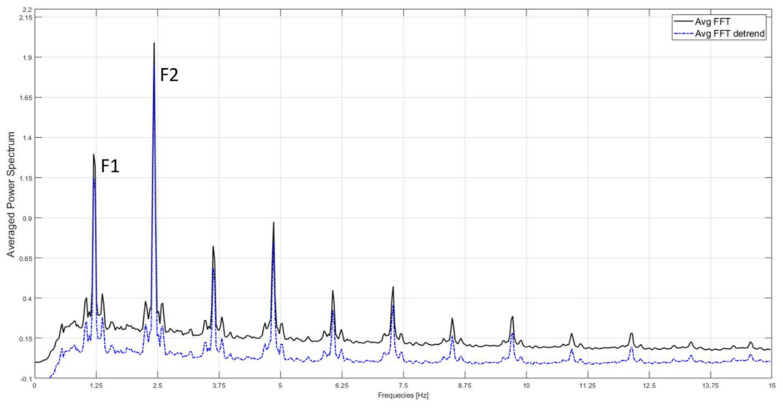
The Averaged Power Spectrum of a representative subject (black curve). The blue dashed line shows the signal after the application of the detrend Matlab function to remove the effect of the flickering noise. The two largest peaks correspond to stride frequency (F1) and cadence (F2).

**Figure 2 sensors-21-02977-f002:**
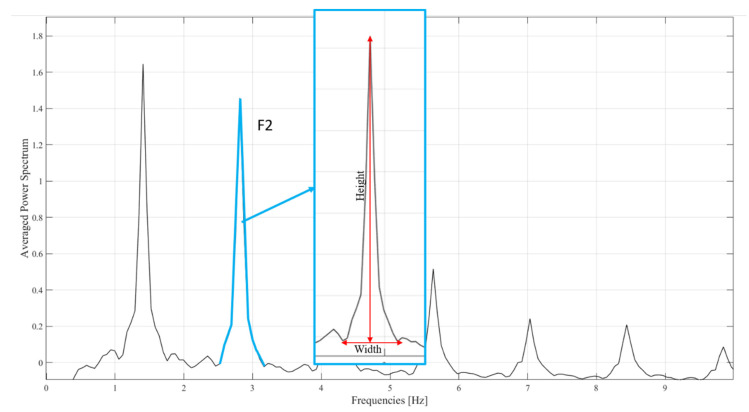
In a representative subject, the Averaged Power Spectrum and the height and width of Frequency 2 (F2), through which the Synchrony Index is calculated.

**Figure 3 sensors-21-02977-f003:**
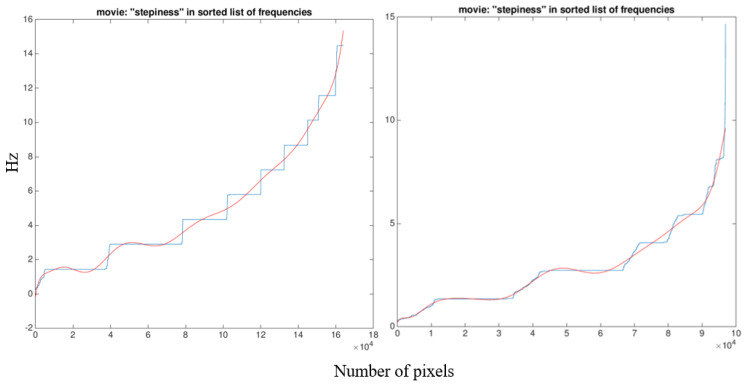
Exported directly from the Graal software output, the left panel shows the frequency-distribution plot in blue and the polynomial fitting function in red of a high-level runner, adapted to the use of treadmill running with a well-coordinated motion pattern. The right panel shows the irregular, worse-quality running gait of a recreational runner, a novice in the use of a treadmill.

**Figure 4 sensors-21-02977-f004:**
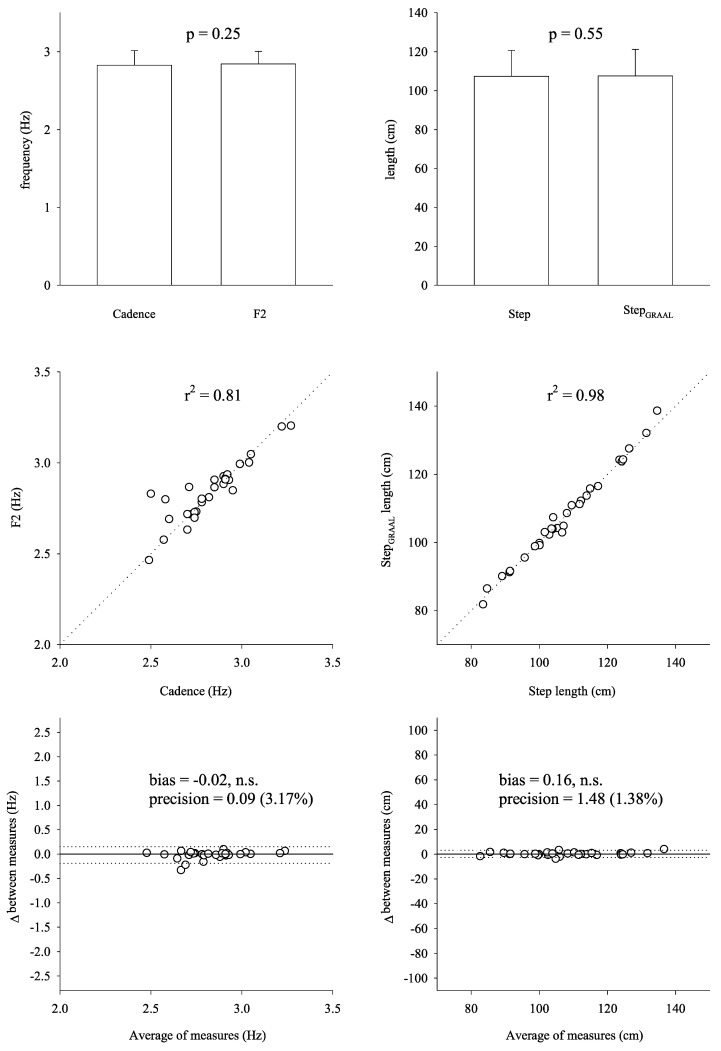
From the top down: histograms, scatterplots, and Bland–Altman plots of cadence vs. F2 frequency (left side) and step length vs. Graal step length comparing measurements by Optogait against those by Graal.

**Table 1 sensors-21-02977-t001:** Characteristics of the population studied. Data are reported as mean (SD).

Variables	Males *n =* 21	Females *n =* 12	Total *n =* 33
Age (years)	40 (9)	41 (12)	40 (10)
Height (cm)	176 (5)	166 (6)	172 (7)
Weight (kg)	72 (8)	56 (6)	67 (10)
BMI	23.1 (1.4)	20.5 (2.0)	22.2 (2.0)

**Table 2 sensors-21-02977-t002:** Values of spatiotemporal gait parameters measured with the Optogait system. Data are reported as mean (SD).

Variables	Males	Females	Total
Speed (km/h)	11.4 (1.3)	10.3 (1.5)	11.0 (1.4)
Step length (cm)	111.4 (11.6)	100.9 (13.7)	108.1 (13.0)
Normalized step length	0.63 (0.06)	0.60 (0.07)	0.62 (0.06)
Height (cm)	175.8 (5.4)	165.7 (5.7)	172.7 (7.2)
Contact time (s)	0.310 (0.035)	0.306 (0.081)	0.309 (0.040)
Flight time (s)	0.046 (0.023)	0.050 (0.034)	0.052 (0.033)
Cadence (steps/min)	170.58 (9.79)	167.52 (14.11)	169.56 (11.26)
Cadence (Hz)	2.82 (0.15)	2.82 (0.19)	2.82 (0.19)
Cadence variability (coefficient of variation of cadence)	3.79 (1.37)	3.88 (1.87)	3.82 (1.50)

**Table 3 sensors-21-02977-t003:** Values of harmony indexes derived from the video analysis method are reported for the entire group (Total) and in sex subgroups (Males, Females). In the last column on the right, the results of the correlation analysis with cadence variability, as measured with Optogait, are reported.

Variables	Males	Females	Total	Correlation with Cadence Variability (vCAD)
Harmony Index	0.21 (0.03)	0.19 (0.02)	0.21 (0.03)	*p* = 0.92, r^2^ = 0.00038
Synchrony index	2.44 (0.96)	2.55 (0.91)	2.32 (0.96)	*p* = 0.87, r^2^ = 0.00122

## Data Availability

The data presented in this study are available on request from the corresponding author. The data are not publicly available due to restriction concerning the privacy policy of the IRCCS Fondazione Don Gnocchi ONLUS.

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
