# Peer review of "Quantitative and Qualitative Running Gait Analysis through an Innovative Video-Based Approach"

_sensors, 2021, doi:10.3390/s21092977_

Round 1

Reviewer 1 Report

The article reports results from a new method called Graal. This is a video-based method for a gait analysis. However, the metodology is not enough explained.

There is not mathematical formulation

How the system detect the leg?

how the system differentiates the cadence between the 2 legs

Why is not presented the test results and the comparison with the Optogait for each person.?

Line 104, "34 healthy individuals". For 21 males + 12 the result is 33

Figure 4 is not centred and too large.

Table 1, the meaning of the parenthesis values are not defined.

Table 2 and line 226, Use gender instead of "sex"

Line 217, eliminate "The" (repeated).

Reviewer 2 Report

The paper at hand presents the results of a running gait analysis, where a video-based approach is compared against a motion capture-based approach. The authors use a video-based system that has been used for walking analysis in the past and test in on running motions.

I have some concerns regarding this work.

  • In the abstract a list of names is given: Graal, Microgate, Bolzano it is not clear what these are. Only later in the paper, it becomes clear that these are companies or software packages that are used. A proper introduction of these tools, and a footnote or citation would be mandatory.
  • Abstract, line 34 to line 39 are a bit confusing. What has been shown in previous works is not part of the abstract. It is fine to report on negative results, but the positive results (hypothesis supported by the data) should be included to the abstract, too.
  • Why are data recorded with a 4K camera when the resolution is set to 640x480px?
  • As this paper is submitted to sensors, I was expecting some technical contribution. However, it seems all technology was developed and published already in [36]. This leaves the contribution of this work to test a working approach for analyzing walking on running data. To better understand the contribution of this work, the authors should clearly list the contributions in the introduction. Especially with some negative results, it is not clear from the manuscript what worked and what did not work.
  • Equation 3 seems to be the first equation in the paper. Please check numbering.
  • Equation 3 is the only equation in the paper. It seems to be cited from [36], like all other technology, too. Nevertheless, it has a typo, $m x s^{-1}$ should be $m \times s^{-1}$.
  • Additionally, some motivation and reasoning for the step length estimation would be helpful.
  • As mentioned before, the paper would benefit from a clearer structure of the contributions and the tested hypothesis. It would be nice to write down the hypothesis that have been tested and report for each if it was rejected on statistically significant level.
  • Line 199: “kmh-1” could be written as “km/h”

In the end, the paper shows that the Graal software can be used as a tool for qualitative analysis of running motions. Nothing more. From my perspective this is to less for a journal paper. I don’t see how a revision would overcome this issue. Thus, I vote for rejecting this paper.

Reviewer 3 Report

Congratulations to the authors for the work entitled "Quantitative and Qualitative Running Gait Analysis through an innovative video-based approach " a very interesting job, the search for low investment clinical resources is necessary to improve the therapeutic approaches. Therefore, I consider that has an interesting approach for publication in Sensors. But, there are some questions of form that should be taken into account prior to consider this article for publication.

I enclose my suggestions for consideration by the authors.

  • The abstract does not make the objective of the work clear, nor is the methodology explained in a synthesized way.
  • Lines 57 and 92. The authors refer to the advantageous use of technologies to evaluate the evolution of the rehabilitation processes and the reincorporation to sports practice. I believe that the authors will share that this type of technology is essential for the evaluation of any clinical intervention, whether it is rehabilitation, treatment or evaluation, for this reason I believe that the authors should reflect its global application and not only to the rehabilitation
  • It is necessary that the authors reflect in the text, or include a table, the description of the population studied .... age, weight, ......
  • Lines 114: “subjectively comfortable speed”, authors should define this concept.
  • The authors state in the text that the type of shoe can interfere with gait , they only consider that they were neutral type. The authors did not take into account the composition of the footwear, mainly the sole? Did you write down these characteristics?, If so, they should reflect it in the text and take it into consideration in the results, otherwise ... Do the authors consider this a limitation of the work? then they should rebled it.
  • In the methodology the authors include several abbreviations that make the reading not very understandable, I recommend that you evaluate this and that you check if the first time you learn they are described.
  • The authors do not cite figures with the same criteria throughout the text.
  • I consider that figure 2 is more correct for a meeting than for this type of document, the authors should assess this aspect.
  • Line 198: “Subjects aged 41(10) years old, weight 69(10) kg and height 174(8) cm.” All the subjects have these characteristics? Are the means? ... this must be clarified ..... it is more important to include the description table of the population studied.
  • The conclusions are not clear.

Round 2

Reviewer 2 Report

The paper at hand presents the results of a running gait analysis, where a video-based approach is compared against a motion capture-based approach. The authors use a video-based system that has been used for walking analysis in the past and test in on running motions.

This revision of the paper is clearly improved compared to the initial submission. It is much better to understand what has been done, how it has been done, and what conclusions we can take home from the described experiments.

Remaining concerns regarding the current version of the paper are:

  • Both systems (the evaluated one, and the gold standard for comparison) are from the same company. In addition, two of the authors are from this company, too. I miss an external controlling factor. Are the results reproducible by other researchers, if they buy the same systems but don’t have access to engineers who know the systems in and out?
  • In equation 1 a clearer write up of the units would increase readability – how do I come from m\timess^-1\times Hz to cm?
  • In line 193 the authors propose a hypothesis, is this accepted or rejected in the paper?
  • The abstract is very long. Maybe shorten it by focusing on the most important aspects. Some sentences might be moved to other parts of the paper or removed from abstract if redundant.
